# Gait Impairment Analysis Using Silhouette Sinogram Signals and Assisted Knowledge Learning

**DOI:** 10.3390/bioengineering11050477

**Published:** 2024-05-10

**Authors:** Mohammed A. Al-masni, Eman N. Marzban, Abobakr Khalil Al-Shamiri, Mugahed A. Al-antari, Maali Ibrahim Alabdulhafith, Noha F. Mahmoud, Nagwan Abdel Samee, Yasser M. Kadah

**Affiliations:** 1Department of Artificial Intelligence and Data Science, College of Software & Convergence Technology, Sejong University, Seoul 05006, Republic of Korea; m.almasani@sejong.ac.kr (M.A.A.-m.); en.mualshz@sejong.ac.kr (M.A.A.-a.); 2Biomedical Engineering Department, Cairo University, Giza 12613, Egypt; eman.marzban@eng1.cu.edu.eg; 3School of Computer Science, University of Southampton Malaysia, Iskandar Puteri 79100, Johor, Malaysia; a.k.n.al-shamiri@soton.ac.uk; 4Department of Information Technology, College of Computer and Information Sciences, Princess Nourah bint Abdulrahman University, Riyadh 11671, Saudi Arabia; mialabdulhafith@pnu.edu.sa; 5Rehabilitation Sciences Department, Health and Rehabilitation Sciences College, Princess Nourah bint Abdulrahman University, Riyadh 11671, Saudi Arabia; nfmahmoud@pnu.edu.sa; 6Electrical and Computer Engineering Department, King Abdulaziz University, Jeddah 22254, Saudi Arabia; ykadah@kau.edu.sa

**Keywords:** gait disorders, gait analysis, deep learning, assisted knowledge learning, silhouette images

## Abstract

The analysis of body motion is a valuable tool in the assessment and diagnosis of gait impairments, particularly those related to neurological disorders. In this study, we propose a novel automated system leveraging artificial intelligence for efficiently analyzing gait impairment from video-recorded images. The proposed methodology encompasses three key aspects. First, we generate a novel one-dimensional representation of each silhouette image, termed a silhouette sinogram, by computing the distance and angle between the centroid and each detected boundary points. This process enables us to effectively utilize relative variations in motion at different angles to detect gait patterns. Second, a one-dimensional convolutional neural network (1D CNN) model is developed and trained by incorporating the consecutive silhouette sinogram signals of silhouette frames to capture spatiotemporal information via assisted knowledge learning. This process allows the network to capture a broader context and temporal dependencies within the gait cycle, enabling a more accurate diagnosis of gait abnormalities. This study conducts training and an evaluation utilizing the publicly accessible INIT GAIT database. Finally, two evaluation schemes are employed: one leveraging individual silhouette frames and the other operating at the subject level, utilizing a majority voting technique. The outcomes of the proposed method showed superior enhancements in gait impairment recognition, with overall F1-scores of 100%, 90.62%, and 77.32% when evaluated based on sinogram signals, and 100%, 100%, and 83.33% when evaluated based on the subject level, for cases involving two, four, and six gait abnormalities, respectively. In conclusion, by comparing the observed locomotor function to a conventional gait pattern often seen in healthy individuals, the recommended approach allows for a quantitative and non-invasive evaluation of locomotion.

## 1. Introduction

Gait is the way an individual walks. It is the locomotion attained through the movement of limbs influenced by the integrated functioning of the musculoskeletal and nervous systems [1]. Any change in this distinctive sequential walking pattern can provide important clues about the state and development of a wide range of underlying health problems [2]. One reliable way to detect declining health symptoms caused by aging, physical dysfunction, or neurodegenerative conditions is by observing gait. Such disease and disorders include, among others, Parkinson’s disease, multiple sclerosis (MS), and strokes [3,4,5,6]. Clinicians analyze the gait patterns of patients using various types of data, such as temporal–spatial, kinematic, kinetic, or electromyography data. By doing so, they can accurately classify these gait patterns through clinically significant groups and develop suitable interventions accordingly [7,8].

When gait pattern and characteristics can be reliably identified in the clinical setting and then tracked and evaluated over time, it opens the door to more effective personalized therapy and more accurate predictive outcome evaluation [2]. It was reported by [9] that understanding clinical rehabilitation and machine learning is crucial, as shown by the evaluation of future research. There has been a recent paradigm shift towards personalized medicine, which considers the uniqueness of each patient. The use of instrumented gait analysis is being extensively employed in the assessment of patients, serving as a valuable tool in clinical decision-making processes and the evaluation of various treatment outcomes [10].

Clinical gait analysis is a multi-dimensional procedure that entails the comprehensive examination and analysis of data derived from various sources [11]. Neurologists and rehabilitation therapists are responsible for conducting various diagnostic examinations to evaluate and manually quantify gait disorders. The development of an automated system capable of identifying individuals with impaired mobility based on their gait patterns holds great promise for enhancing health care and various other applications related to gait analysis [12,13,14,15,16]. Physical examinations, the collection of patient history, and three-dimensional motion analysis collectively provide valuable data pertaining to the patient’s gait impairments. Identifying patterns and trends within this extensive dataset can pose challenges. Nevertheless, the interpretation of clinical gait analysis results and subsequent treatment decisions are predominantly based on subjective pattern recognition by clinicians. Hence, the development of a method to accurately classify gait data could be beneficial in identifying cohorts of individuals sharing widespread underlying clinical conditions.

The utilization of automated recognition techniques for human gait patterns enables several valuable applications. Firstly, it facilitates a quantitative and non-invasive diagnosis of locomotion by comparing the observed locomotor function to a standard gait pattern associated with healthy individuals [17,18]. Secondly, it facilitates the identification of personalized gait training tasks by automatically adapting assistance based on the individual’s recognized motor function [15]. Thirdly, it aids in the planning of potential treatment strategies tailored to the specific needs of the user, considering their previously identified gait impairment through automated recognition approaches [19]. Lastly, it enables the measurement and description of the progress made in gait treatment by assessing the individual’s gait patterns during initial and follow-up sessions [7,20]. Moreover, these automated systems utilized in clinical gait analysis represent an impartial method for the extensive manipulation of gait data, and they offer enhanced efficiency and cost effectiveness compared to the traditional procedures commonly employed by clinicians [21].

The existing research on gait analysis can be broken down into two groups: model-based and appearance-based techniques. In order to obtain reliable pose parameters, model-based approaches often use a motion capture device or wearable sensors. However, model-based techniques require controlled laboratory conditions and subject cooperation; subjects must wear specialized equipment or gadgets in order to obtain accurate measurements of the trajectories of their joints and other moving body parts. Additionally, they typically endure high computation costs and poor pose estimation quality. On the other hand, appearance-based solutions employ captured videos directly; this eliminates the requirement for individuals to wear any specialized equipment. The method relies on analyzing a person’s gait in order to determine things like their gender and verify their identity; the gait energy image (GEI) [22] is one such feature that has the potential to be employed for gait classification due to its strong authentication performance for individuals based on gait. In addition to its other benefits, the GEI does not necessitate high-quality silhouette extraction and is hence highly resilient against the noise that is invariably present in extracted silhouette images. Due to its definition as the mean of successive silhouettes over a certain walking period, the GEI is well suited to encoding the shape of people and thus could prove useful for verification purposes. Despite the remarkable performance exhibited by the GEI feature in gait classification, it relies on prior knowledge of gait cycle sequences and requires the use of a two-dimensional classification model. To solve this problem, the proposed study seeks to investigate the effectiveness of alternative image features, namely silhouette sinograms, which bear resemblance to the GEI feature, in detecting gait impairments. To utilize these features, a 1D classification model that does not rely on gait cycle information is required. The following are the novel contributions achieved through this study.

Novel silhouette sinogram signals are generated for each silhouette image by computing the angle and distance between the centroid and all boundary points, facilitating the extraction of prominent representations of relative variations in pathological gait patterns.We develop a 1D CNN model that can integrate multiple silhouette sinogram signals and train them together to enhance the network’s understanding of gait abnormalities through assisted knowledge learning.We analyze the influence of varying k-assisted silhouette sinogram signals, used as prior knowledge and fed into the network (k= 1, 10, 20, and 30). This work also conducts comprehensive experiments encompassing three distinct sizes of gait abnormalities, categorized into n=2,4, and 6 classes.This work employs two evaluation methodologies: one based on individual silhouette frames and another operating at the subject level using a majority voting approach.

Quantitative gait analysis has been around for over 30 years, although it is still mostly used in labs and not much in clinical practice. This is mainly because clinical teams have varying levels of training, experience, and preferences, and because typical gait labs include expensive and heavy equipment, complicated protocols, and data management and analysis. Despite evidence of the decreased efficacy of quantifying gait, observational gait and qualitative measures are still employed extensively in clinics [2]. We believe that this gait representation method has not been proposed in the literature before. It offers the opportunity to utilize low-cost camera-based gait data collection methods to obtain detailed information about the movement of different parts of the body, similar to what sensor-based methods offer. Furthermore, the computational complexity is very modest, allowing for real-time implementation.

The rest of this paper is outlined as follows: Section 2 introduces related works and investigations for gait impairment analysis. Section 3 describes our proposed methodology, including the generation of silhouette sinogram signals and the developed 1D CNN model. Section 4 presents the experimental results and comparisons. Section 5 and Section 6 discuss the findings and draw conclusions, respectively.

## 2. Related Works

The application of movement analysis has demonstrated its utility in comprehending the underlying mechanisms of various pediatric diseases and enhancing their subsequent monitoring and evaluation [23]. The utilization of a particular intervention in the management of cerebral palsy has experienced significant prevalence in recent years. According to Hebda-Boon et al. [24], the implementation of this intervention within hospital settings has led to a decrease in the number of surgical treatment recommendations. Additionally, it has been employed to enhance rehabilitation in diverse pediatric conditions by emphasizing the osteoarticular hazards caused by excessive trunk tilt motions [25]. Furthermore, the integration of motion analysis with X-ray imaging has been employed to enhance our comprehension of the correlation between alterations in skeletal structure and corresponding modifications in individuals with cerebral palsy [23] as well as those with diverse postural abnormalities [26].

We can categorize the current body of literature on gait analysis into two distinct categories: model-based approaches and appearance-based approaches. Typically, model-based approaches employ either a motion capture system [27,28,29,30] or wearable sensors [27,31,32,33,34,35] to accurately capture pose parameters. In their study, Hallemans et al. [36] employed a three-dimensional motion capture system to assess head orientation, stride length, and trunk flexion. Their objective was to investigate the potential impact of impaired vision on the dynamic stability of walking. Several model-based studies conducted between 2015 and 2020 have used camera-recorded movies to estimate pose parameters. These studies achieved this by employing fitting and tracking techniques on various body components. In their study, Khan et al. [37] utilized a skeletal model to analyze binary human silhouettes. This analysis allowed them to determine the posture lean and stride cycles of individuals, with the aim of detecting Parkinsonian gait. Costin et al. [38] have proposed a novel automated image processing technique for the assessment of human gait in patients undergoing rehabilitation. The present study assessed the atypical gait patterns exhibited by patients as a result of lower limb injuries. They used the MoveNet Thunder model to determine the positions of joints and key points in a human’s body based on video data. The magnitudes as well as variations in the angles resulting from the symmetrical articulations of the human body are assessed in different frames, following the identification of specific body parts. The purpose of this evaluation is to analyze and monitor these angles throughout the rehabilitation process. The concept of a rehabilitation degree was established and quantified as the mean percentage of the pertinent angles of the body segments relative to the domains of normality for those angles. Based on the encouraging outcomes obtained from the utilization of these two model-based methodologies, it is affirmed that the identification of disordered gait necessitates the consideration of crucial factors such as the walking posture, temporal cue, and stability property. Indeed, model-based methodologies have the capability to accurately quantify the trajectories of mobile joints and bodily elements. However, their implementation necessitates controlled laboratory conditions and the active participation of the subjects, who are required to don specialized attire or apparatus. Moreover, it is common for these systems to exhibit subpar accuracy in estimating poses and require significant computational resources. Therefore, these methods are not suitable for the scheme we aim to accomplish.

In contrast, appearance-based methodologies rely on the direct utilization of recorded videos, thereby eliminating the necessity for subjects to use any specialized equipment. Chen et al. [13,14] employed binary silhouettes derived from color images as a means of differentiating individuals diagnosed with Parkinson’s disease from those without the condition. Their study used binary images directly for classification purposes, making the results significantly dependent on the caliber of each silhouette. Previous studies [22,39,40] have attempted to analyze gait observations in order to identify certain characteristics that can be utilized for the purposes of gender recognition and human authentication. One frequently employed feature in gait analysis is the gait energy image, which has gained recognition for its ability to effectively authenticate individuals based on their gait patterns. One additional benefit of the GEI method is its ability to handle noise present in extracted silhouette images without the need for high-quality silhouette extraction. Given that GEIs are characterized as the means of consecutive silhouettes within a walking interval, it possesses a strong capability to accurately represent the physical form of individuals. Consequently, it is reasonable to assert that the GEI is a highly efficient method for the identification of pathological gait impairments from 2D camera-based and video-recorded images.

Gait recognition is a contemporary biometric technique that has gained prominence for its potential applications in security. It is characterized by its unobtrusive learning method. Bakchy et al. [41] employed the GEI feature for the purpose of gait recognition in security-related applications. The GEI effectively retains both the dynamic and static characteristics inherent in a gait sequence. The static information pertaining to the human body encompasses its physical appearance and shape, while the dynamic information encompasses fluctuations in frequency and phase. Nevertheless, the temporal aspect that normalizes each silhouette within the GEI is not taken into account. In response to this issue, Luo et al. [42] suggested the cumulative frame difference energy image, which can reflect the time properties. Subsequently, gait recognition was achieved by employing the nearest neighbor classifier, which relied on the Euclidean distance metric. The experimental findings indicate that the proposed algorithm exhibits superior performance compared to the hybrid GEI and 2D-PCA methods, while also satisfying the real-time constraints. The GEI feature has demonstrated remarkable efficacy in gait classification, prompting us to improve the accuracy of current classification models by harnessing the capabilities of convolutional neural networks. One potential approach to achieve this objective involves employing sinogram images as an alternative to GEIs.

In the work of Chen et al. [13], they used an LDA as a feature extractor, followed by a minimum distance classifier (MDC) to differentiate patients with Parkinson’s from normal subjects using video sequences as the input, and reached an overall accuracy of 95.49%. Their dataset comprised seven healthy subject and seven PD patients having video sequences of around 2.57 ± 0.5 min. In 2018, Verlekar et al. [43] used SVM to classify four gait patterns using the INIT. The involved features were body-related features such as the torso orientation and the shift in the center of gravity, and feet-related features such as the step length and the ratio between the foot flat period to the stance period. The parameters involved in their system involved the entire video sequence for both the training and the testing phases, and they reached an accuracy of 98.8%. In 2021, Albuquerque et al. [44] created their own simulated dataset named GAIT-IT and trained their CNN on it. They used GEIs and achieved an accuracy of 93.4% and 89.8% when testing on GAIT-IT and GAIT-IST [45], respectively. They included five gait types for classification purposes, namely, normal, scissor, spastic, steppage, and propulsive, which differ from the pathologies included in the INIT dataset. The same team, in their published work [46], achieved an accuracy of 96.5% and 91.4% when testing on GAIT-IT [44] and GAIT-IST [45], respectively. In this work, they selected nine key frames per gait cycle, then used a pretrained CNN for spatial feature extraction and a bidirectional LSTM for temporal feature extraction. Finally, the classification was performed by two fully connected neural network layers. In the studies of [13,43,44,46], the accuracy was the sole evaluation metric computed.

The current state of the art for predicting gait impairment in medical applications lacks advanced deep learning methods such as Federated Learning (FL). FL has emerged as a promising technique in the field of machine learning, particularly within the realm of mobile health (mHealth) [47]. Unlike conventional machine learning methods, FL enables collaborative model training across multiple data sources without requiring direct access to individual datasets. This approach is especially beneficial in the context of mHealth, where medical data are often fragmented across isolated silos, and privacy concerns pose significant barriers to data sharing. As a result, FL holds great potential for advancing research and innovation in the field of mHealth, offering a privacy-preserving solution for leveraging distributed data sources effectively. This approach, explored by Zhang et al. [48,49] for tasks like predicting the remaining useful life, holds promise for gait impairment prediction as well.

## 3. Materials and Methods

### 3.1. Dataset

Gait difficulties refer to any deviation from the standard walking or gait pattern. Neurological etiologies exhibit a higher prevalence compared to non-neurological etiologies. There are a multitude of diseases that impact on both the central and peripheral nervous systems, resulting in subsequent alterations to an individual’s gait. Gait disturbances may arise as a result of subtle alterations in the interactions between these two systems. The examination should encompass the following aspects: Posture, the act of being in an upright position, with one’s weight supported by one’s feet, refers to the position and alignment of the body, particularly in relation to the spine and limbs. It encompasses various aspects, such as narrow or wide stances. Additionally, the examination involves the evaluation of the walking process and the measurement of the step length. The gait cycle consists of two primary phases: stance, during which a specific foot is in contact with the ground, and swing, during which the same foot is no longer in contact with the ground and is propelling forward. In the context of typical human locomotion, it is anticipated that the stance and swing phases will account for approximately 62% and 38% of a complete gait cycle; accordingly, Figure 1 depicts the distribution of this phenomenon, as observed from the perspective of the right limb, throughout an entire gait cycle. In this research, a rehabilitation specialist examined several video-based indicators meant to distinguish between gait phases and instances.

In this investigation, we made use of an already existing data collection called the INIT Gait database [5,50]. This dataset is made up of sequences of high-quality binary silhouettes that were extracted from RGB movies that were recorded at the specialized studio LABCOM at University Jaume I. LABCOM is a part of the audiovisual facilities at the university. A group of ten individuals, consisting of nine males and one female, were recruited as healthy volunteers to participate in a study including the simulation of several aberrant gait styles on a green chroma background. The utilization of a consistent background aided the process of converting the frames into binary form and extracting silhouettes of superior quality. This, in turn, decreased the level of uncertainty when assessing the precision of features. A total of seven different gait types were mimicked, involving modifications to both limb movement and overall body posture. The individuals in question draw inspiration from problematic gait patterns that are commonly associated with specific neurological disorders, such as Parkinson’s disease. Additionally, a novel eighth style characterized by natural and uncontrived movement was incorporated. Each participant was recorded twice for each gait pattern, and all sequences were obtained from a lateral perspective, enabling a more detailed description of limb action and body position. Hence, a total of 160 sequences were captured, including a collective count of 41,104 silhouette images. Each of these images has a frame size of 400 × 800 pixels. The following table, Table 1, provides a summary of the gait styles contained in the INIT Gait database.

### 3.2. Proposed Framework

This study proposes a new pipeline for gait impairment analysis, consisting of three main stages. Firstly, silhouette sinogram signals are generated utilizing a novel angular centroid distance algorithm. Subsequently, a one-dimensional convolutional neural network (1D CNN) model is developed and trained by integrating shared information through assisted knowledge learning. Finally, two evaluation schemes are employed: one based on individual silhouette frames and another based on the subject level, utilizing a majority voting approach. The diagram scheme of the proposed framework is illustrated in Figure 2.

#### 3.2.1. Generation of Silhouette Sinogram Signals

(1)Image Cropping

The first step is to segment and crop all binary silhouettes in each subject’s gait cycle so that they fit within the same image size. This is achieved by projecting the original silhouettes in the horizontal and vertical directions, then searching for nonzero elements within these projections to find the start and end points as the minimum and maximum of nonzero element positions. Given an initial binary silhouette image n in the sequence $S_{n}(x,y)$, the procedure can be expressed mathematically as follows:(1)$$S_{nx}\left(y\right)=\sum_{x}S(x,y)\;\mathrm{and}\;S_{ny}\left(x\right)=\sum_{y}S(x,y).$$

Then, the cropped image $S_{n}^{c}\left(x,y\right)$ can be obtained as the subset of rows and columns where the projections in the *x* and *y* directions are both nonzero, such that
(2)$$S_{n}^{c}\left(x,y\right)={\left\{S_{n}\left(x,y\right)\right\}}_{{\;S}_{nx}\left(y\right)>0\;and{\;\;S}_{ny}\left(x\right)>0}.$$

The cropped image sizes $N_{nx}$ and $N_{ny}$ are obtained as the cardinality of all nonzero elements in the projection vectors $S_{ny}\left(x\right)$ and $S_{nx}\left(y\right)$, which can be calculated as the 0-norm of each:(3)$$N_{nx}={\left\|S_{ny}\left(x\right)\right\|}_{0},\;N_{ny}={\left\|S_{nx}\left(y\right)\right\|}_{0}.$$

Each binary silhouette is cropped in both the horizontal and vertical directions to contain only the points from the start and end point locations. This results in a sequence of binary silhouette images of variable horizontal and vertical sizes. In order to make all of them have the same size, all images are zero-padded equally from both directions to the maximum vertical and horizontal sizes of all nonuniformly sized cropped images to obtain uniformly sized cropped binary silhouette images. The common binary silhouette sizes in a particular sequence are obtained as follows:(4)$$N_{x}=\underset{n}{\max}\left\{N_{nx}\right\},\;\;N_{y}=\underset{n}{\max}\left\{N_{ny}\right\}.$$

The result of zero-padding all binary silhouette images in the sequence to the common sizes $N_{x}$ and $N_{y}$ is termed $S_{n}^{cz}\left(x,y\right)$ for the image *n* in the sequence. This will ensure consistency in the subsequent movement assessment steps for the different silhouettes in the gait sequence.

(2)Centroid and Boundary Calculations

For each gait sequence, the centroid location $\left(c_{nx},\;c_{ny}\right)$ of each of the uniformly sized cropped binary silhouette images $S_{n}^{cz}\left(x,y\right)$ is calculated as the location of the center of mass of each assuming uniform mass density everywhere in the silhouette region. This can be expressed mathematically as
(5)$$c_{nx}=\frac{\sum_{y}\sum_{x}x\;S_{n}^{cz}\left(x,y\right)}{\sum_{y}\sum_{x}\;S_{n}^{cz}\left(x,y\right)},\;c_{ny}=\frac{\sum_{y}\sum_{x}y\;S_{n}^{cz}\left(x,y\right)}{\sum_{y}\sum_{x}\;S_{n}^{cz}\left(x,y\right)}.$$

Then, uniformly sized cropped binary silhouette images undergo a boundary tracing operation to obtain an ordered sequence of points that represent the silhouette boundary. This is more sophisticated than edge detection in that the boundary is given as an ordered sequence of boundary image points rather than a binary contour image, particularly with complex silhouette region shapes. This can be performed using a number of existing approaches such as pixel-following- or vertex-following-based methods, where the basic assumptions of having input binary images that are zero-padded with no boundary points at the edge of the image matrix. In this study, the boundary tracing procedure was implemented based on the Moore neighbor tracing algorithm modified by Jacob’s stopping criteria [51]. This starts with a boundary point (preferably the uppermost and leftmost one) and proceeds to examine its eight neighbors one by one in the same direction (for example, clockwise) starting from a neighbor that has a zero value (lies in the background) and selecting the first neighbor that has a one value (lies within the silhouette) to be the next point in the boundary and sequence. This process is repeated until one of Jacob’s stopping criteria is met whereby the sequence of boundary points reaches the starting point. The result of this boundary detection operation is a list of boundary points’ horizontal and vertical locations $\left\{{{\vec{b}}_{nx},\;\vec{b}}_{ny}\right\}$ for each cropped binary silhouette image $S_{n}^{cz}\left(x,y\right)$. It should be noted that the size of such boundary sequences will vary across different images within the same sequence.

(3)Silhouette Sinogram Calculations

In this part, the detected boundaries $\left\{{{\vec{b}}_{nx},\;\vec{b}}_{ny}\right\}$ of uniformly sized cropped binary silhouettes $S_{n}^{cz}\left(x,y\right)$ along with their centroids $\left(c_{nx},\;c_{ny}\right)$ are used to generate a novel one-dimensional representation of each silhouette, which we will term a silhouette sinogram. This is achieved by computing the vectors of distances and angles $\left\{{{\vec{D}}_{n},\;\vec{A}}_{n}\right\}$ between the centroid and each of the silhouette boundary points, as shown in Figure 3. The mathematical representation of these calculations for the point *m* in the boundary sequence are as follows:(6)$$D_{nm}={\left\|\left[\begin{array}{c} c_{nx}\\ c_{ny} \end{array}\right]-\left[\begin{array}{c} b_{nmx}\\ b_{nmy} \end{array}\right]\right\|}_{2},\;A_{nm}=\tan^{-1}\left(\frac{b_{nmy}-c_{ny}}{b_{nmx}-c\_nx}\right).$$

It should be noted that this representation is a generalization of the so-called signature of the 2D boundary [51], where the angles included in this representation are nonuniformly sampled in this work. We observe that this distance is, in essence, a projection of the silhouette in the angle that the line between each boundary point and the centroid makes with the horizontal direction. Such a one-dimensional representation of the two-dimensional silhouette in terms of projections bears similarity to the sinograms encountered in computed tomography. Hence, such a distance versus angle representation will be termed the silhouette sinogram in reference to such similarity.

Since the result of the above step is a set of nonuniformly sampled angles and their corresponding distances, issues related to the multiplicity of boundary points at the same angle as well as the comparison of the distance profiles across different silhouettes in the sequence may arise. Given that the same angle can have multiple distances, such as those corresponding to the proximal and distal boundary points of an extended arm, the algorithm should be able to resolve such multiplicity occurrences when detected to have only one distance value for each angle. This is achieved by sorting the set of angles and their corresponding distances by the angle values, the detection of multiplicity for each angle, and selecting the maximum distance for each angle from the different distance values. This ensures that the maximum extension information is kept given its greater relevance to the classification. The above steps can be expressed mathematically as follows:(7)$$\left\{A_{nm}^{\prime }\right\}=\underset{m}{\mathrm{Sort}}\left(\left\{A_{nm}^{\prime }\right\}\right).$$
(8)$$D_{nm}^{\prime }=\max\left(\left\{D_{nm}\right\}\;\forall \;D_{nm}\;with\;similar\;A_{nm}\right).$$

Subsequently, the set of angles and their corresponding distances now contain consistent entries that are repeated whenever multiplicity is encountered. Therefore, this set should be reduced to include only distinctive angle/distance pairs $\left\{{{\vec{D\prime }}_{n},\;\vec{A\prime }}_{n}\right\}$, such that
(9)$${{\vec{D^{\prime }}}_{n}={\left\{A_{nm}^{\prime }\right\}}_{\neq }\;\;\;,\;\;\;\vec{A^{\prime }}}_{n}={\left\{D_{nm}^{\prime }\right\}}_{\neq }.$$

Here, the common mathematical notation of ${\left\{\;.\right\}}_{\neq }$ is used to express the selection of unique elements within the enclosed set.

Furthermore, given that the resultant angle values are not uniformly sampled and that such sampling differs from one silhouette to another in the same sequence, resampling of the distance versus angle data is performed. The outcome of this resampling allows the final silhouette sinogram to be obtained as the distance versus angle data uniformly sampled with a specified resolution ∆A from 0 to 360 degrees. This can be achieved using 1D interpolation using linear or cubic interpolation methods to obtain the resampled sinogram signal ${\vec{D^{r}}}_{n}$. Here, linear interpolation was used. This is mathematically expressed as follows:(10)$${\vec{D^{r}}}_{n}=\underset{\;\vec{A_{n}^{r}}=\left\{0,\;∆A,\;2∆A,\;\ldots \right\}}{\mathrm{Resample}}\left(\left\{{{\vec{D^{\prime }}}_{n},\;\vec{A^{\prime }}}_{n}\right\}\right).$$

In this study, a 2-degree resolution was found to be robust in reducing small variations in close points while being adequate for maintaining the details of sinogram representation. Furthermore, the selection ensures that the resampling procedure will not encounter an aliasing problem given that the original density of the sampling will be more than or equal to the desired uniform sampling.

Once the silhouette sinograms are obtained from the gait sequence, we have a set of signals with each representing the distance versus angle profile for a silhouette at a certain point in time. The silhouette sinograms can be displayed as overlapping thin line plots or as an image where the distance is coded as intensity. A diagram showing the process of obtaining the silhouette sinograms is shown in Figure 3. It should be noted that the silhouette sinograms show marked differences between the different gait classifications as shown in Figure 4 and hence have significant potential for use in computer-aided diagnosis.

It can be observed that the variations in such silhouette sinograms along the sequence of images in the gait cycle provide data that are similar to those of the more expensive sensor-based gait data acquisition setups. It should also be noted that silhouette sinograms can be normalized such that their variations across subjects of different height or body shape can be minimized whereas relative variations in motion at different angles are reliably used to detect gait classification.

#### 3.2.2. Deep Learning Network Architecture

The main objective of the proposed deep learning network is to effectively differentiate between various types of gait abnormalities. We have adapted and introduced a 1D CNN architecture, inspired by the success of 2D CNNs in various image classification tasks in both natural and medical image domains. This architecture aims to learn and extract distinctive features of pathological gaits associated with each specific gait pattern. This adaptation allows for a more accurate assessment and evaluation of gait impairment, ultimately enhancing the analysis of gait-related abnormalities.

The aim behind the design of a 1D CNN structure is to seamlessly integrate it with the input data, denoted as $x_{i}$, which was generated in the aforementioned section as 1D silhouette sinogram signal information for each individual ith silhouette image. The proposed network architecture involves an encoder comprising four convolutional resolution blocks of varying sizes and eventually attached to fully connected neural networks (FC-NNs). Within each block, we employ two consecutive 1D convolutional layers, accompanied by batch normalization and rectified linear unit (ReLU) activation. In place of traditional subsampling pooling layers between these blocks, we employ stride convolutions with a value of 2. This approach promotes the further learning of pooling operations while enhancing the overall stability of the network [52]. Specifically, our convolutional filters consist of feature maps with dimensions of 32, 64, 128, and 256, kernel sizes of 7, 5, 5, and 5, and stride values of 1 for the four convolutional blocks, respectively. To further regularize the network training and avoid overfitting, a dropout layer with a probability of 0.3 is incorporated at the top of each block. Note that these convolutional encoder layers are responsible for automatically extracting and learning gait features directly from the input silhouette sinogram signals. The structural overview of the proposed 1D CNN is visually depicted in Figure 5.

Afterwards, these representation patterns are fed into a global average pooling (GAP) layer, which computes a single piece of spatial information from each feature map. This pooling operation aids in preventing overfitting by eliminating the need to optimize additional parameters, resulting in a more robust and generalized model. The extracted prominent features are subsequently passed into three dense FC-NN layers with 64, 16, and n neurons, respectively. Here, the value of n corresponds to the number of classes representing the different gait impairment abnormalities under investigation. The final dense layer is connected to the Softmax activation function, facilitating the classification process. It is noteworthy that this paper investigates three distinct sizes of abnormalities, which are categorized into n=2,4, and 6 classes. In other words, three separate networks are trained and tested, differing solely in the number of classes specified in the last dense FC-NN layer.

One significant contribution of the proposed network architecture lies in its capability to acquire and learn additional assisted knowledge from the consecutive frames or silhouettes of each subject’s data. This aspect is explained in detail in the following subsection, highlighting the network’s capacity to leverage temporal information for enhanced learning and analysis.

#### 3.2.3. Assisted Knowledge Learning

The proposed network is initiated by incorporating the subsequent silhouette sinogram signals of silhouette frames along with the current $x_{i}$ input. This integration was achieved through the re-design of the 1D CNN, enabling the passage of multiple inputs into the network instead of a single signal. The objective of this process is to emulate the provision of essential knowledge about the gait cycle, thereby enhancing the network’s ability to discern gait abnormalities more effectively. Specifically, multiple k-assisted silhouette sinogram signals are utilized as prior knowledge, derived from the same subject exhibiting a similar pattern. This assistance aids the network in making more accurate diagnostic decisions regarding the main $x_{i}$ input. It is of note that this process represents a significant improvement compared to using only a single sinogram signal $x_{i}$.

The process of collaboratively assisted knowledge learning is accomplished through the sharing of intelligence from subsequent $x_{i}$ sinogram signals. In regard to implementation, this process involves performing the following mathematic computations:(11)$$X=Concat\langle \Phi \left(x_{1}\right),\Phi \left(x_{2}\right),\ldots ,\Phi \left(x_{k}\right)\rangle ,$$
where $\Phi \left(x_{i}\right)$ represents the feature extractor for each consecutive sinogram signal, utilizing a separate convolutional block with a feature map of 32 and a kernel size of 7. $Concat\langle \cdot \rangle$ refers to the concatenation process of all the generated features derived from the shared information across k signals. Consequently, this collective knowledge, denoted as X, is then utilized as an input for the designed 1D CNN model. In this study, we investigate multiple values of k, specifically, k = 10, 20, or 30.

By incorporating this shared information into the model, we aim to leverage the accumulated insights from multiple silhouette sinogram signals to enhance the network’s understanding of gait abnormalities. This collaborative assisted knowledge learning approach allows the network to capture a broader context and temporal dependencies within the gait cycle, enabling the more accurate diagnosis and classification of abnormalities. Through our investigation of various values for k, we seek to determine the optimal amount of shared information that yields the best performance in distinguishing gait abnormalities. This process is applicable to all input sinogram signals within both the training and testing datasets. Thus, the network is equipped with a single essential sinogram signal to evaluate, accompanied by k−1 supportive knowledge derived from the subsequent silhouette frames. This design results in a multiple input single output model, wherein at any given time, the network generates a single decision pertaining to the essential input signal. By adopting this approach, we ensure that the network’s classification decisions are based on a comprehensive understanding of the gait cycle, leveraging both the individual silhouette sinogram signals and the collective information provided by the subsequent frames. Figure 5 illustrates the concept of assisted knowledge learning.

#### 3.2.4. Technical Specifications

This study employs a supervised learning approach to train the proposed 1D CNN. This methodology involves tuning and optimizing the network parameters through forward and backward propagation during training epochs. The optimization process compares the early predictions of the network with reference true labels. The implementation details of the proposed work are outlined as follows. A learning rate of 0.003 is initially set, and then exponentially decreased by a factor of 10 until the 50th epoch. To ensure the inclusion of all frames or silhouette images of one gait cycle, a batch size of 50 is utilized. The networks are optimized using the Adam optimizer with the categorical cross-entropy loss function $\left(L_{CE}\right)$. The $L_{CE}$ is defined as
(12)$$L_{CE}=-\sum_{i=1}^{i=N}y_{i}\cdot log\left({\hat{y}}_{i}\right),$$
where $y_{i}$ and ${\hat{y}}_{i}$ are the true label and predicted results from the proposed network, respectively. The summation over N represents the integration of cross-entropy loss for all classes.

The system implementation for this work was carried out on a PC equipped with a Cuda-enabled GPU, specifically the NVIDIA GeForce RTX 3080, and had 64 GB of RAM. The implementation of this work was performed using Python programming language version 3.7.10, along with the Tensorflow framework and the Keras library.

#### 3.2.5. Evaluation Measures

Quantitative assessments were conducted on an unseen test set, which consisted of 20% of the entire INIT Gait dataset. The testing set comprised two subjects, each with two different experiments encompassing various gait pattern abnormalities. Therefore, the total number of experiments in the testing set amounted to four. This paper employed two distinct evaluation schemes. The first scheme involved comparing each predicted sinogram signal, which corresponds to an individual silhouette image, against its corresponding ground-truth label. The second evaluation scheme operated at the subject level, using a majority voting method to consider all silhouette sinogram signals (or silhouette images). Both evaluations involved conducting experimental studies with varying numbers of target classes, specifically, n= 2, 4, or 6. We trained the network as a binary classification system in the two-class problem, which distinguished between normal gait patterns (NM) and severe gait impairment affecting the full body (FB). For the four-class problem, two additional gait patterns were included, namely, half motion of the left leg (LL) and half motion of the right leg (RL). The six-class problem included the four aforementioned gait impairments, along with motionlessness in the upper limbs of the left arm (LA-0) and right arm (RA-0).

We quantitatively evaluated all these experiments using metrics like sensitivity, specificity, accuracy, and F1-score. Additionally, we provided a confusion matrix to visually depict the ratio of correctly predicted classes to false predictions, thereby providing an intuitive representation of the classification performance.

## 4. Results and Experiments

### 4.1. Gait Analysis Results

In this work, a 1D-CNN model was implemented for the classification purposes of gait pathologies. The test set comprised two subjects, each of which had two sequences for each of the five anomalies as well as the normal unimpaired sequences (NM). The total number of frames was 818, 1292, 1258, 843, 842, and 2616, corresponding to NM, LL, RL, LA-0, RA-0, and FB, respectively.

In this work, three classification problems were addressed: the binary classification of NM versus FB (n = 2); the classification of NM, FB, LL, and RL (n = 4); and the classification of NM, FB, LL, RL, LA-0, and RA-0 (n = 6). In addition, the impact of a different number of signals or images fed to the network was studied (k = 1, 10, 20, and 30).

For the first classification problem (n = 2 and k = 1), NM vs. FB, one image from the sequence was sufficient to achieve perfect classification results: 100% accuracy for all of the 818 NM and 2616 FB frames. Accordingly, there was neither a need to explore the impact of larger values of k, nor a need to evaluate based on a subject’s complete sequence. Figure 6 displays the confusion matrices of all the other problems using both evaluation schemes. To analyze the proportion of correct classifications vs. misclassification with respect to each class, their proportions were calculated and are shown in Figure 7 and Figure 8 for n = 4 and n = 6, respectively.

As can be seen, for all the values of n and k, no misclassifications occurred with FB deformation, where a subject suffered from severe gait cycle impairment featuring short steps, feet subtly touching the ground, and bending knees, chest, and head [5].

Table 2 shows the evaluation metrics for the aforementioned classification problems, weighted by the total number of frames per class. The best results, those having the highest F1-score, are highlighted; hence, testing the network with 20 and 30 images for the four-class and the six-class problems, respectively, is recommended to attain the best performance. The accuracy is computed as the percentage of the correct classifications divided by the entire population, as per Kelleher et al. [53]; it is also equivalent to the weighted sensitivity, and they recommended this measure of accuracy over the weighted accuracy for the imbalanced datasets.

Table 2 depicts portions of the sequences fed to the network along with their predictions. The results are also displayed in Appendix A, where all the evaluation metrics were computed per class individually. Table 3 displays the class metrics for the suggested values of k—achieving the highest weighted F1-score—that were 20 and 30 for n = 4 and n = 6, respectively. The highest F1-score was computed for FB, followed by RL, LL, NM, RA, and lastly, LA. On the other hand, the highest accuracy was computed for FB, followed by NM, LL, RL, LA, and lastly, RA. These values were computed when the class in the header of the table was regarded as the positive class and all the remaining classes were the negative class.

As mentioned above, all the FB frames were successfully classified for all the values of n and k; a sample of a sequence is shown in Table 4. In addition, the samples for an NM subject as well as an LL subject are shown in Table 4 with different values of n and k.

Figure 9 shows the detailed results of the four-class problem on the test set. In this figure, the *x*-axis denotes the subject ID along with the sequence number; e.g., NM1_Seq1 denotes the first sequence for the first subject with normal gait, and LL1_Seq2 denotes the second sequence of the first subject representing problems with the left leg, and so on. All the subjects’ sequences are displayed except the four FB sequences since they yielded a 100% accuracy (for both n = 4 and n = 6). This bar chart portrays the proportions of labels for each sequence with respect to the number of frames. For example, regarding the entire sequence of frames for one of the two experiments of the first subject, when k=1, frames predicted as NM, LL, and RL comprised 26%, 34%, and 40%, respectively; when k=10, frames predicted as NM, LL, and RL comprised 71%, 11%, and 18%, respectively; when k=20, frames predicted as NM, LL, and RL comprised 79%, 7%, and 14%, respectively; and, lastly, when k=30, frames predicted as NM, LL, and RL comprised 93%, 3%, and 4%, respectively. For all the values of k and n, no misclassification with FB was encountered. All of these values are visualized as the first four bars and so on for the rest of the subjects. In a similar fashion, Figure 10 displays the detailed results of the six-class problem on the test set.

### 4.2. Comparative Experiments

This study utilizes a deep learning method to identify abnormal gait patterns associated with neurological conditions like Parkinson’s disease, multiple sclerosis, and strokes. Our focus is on gait classification, and as shown in Table 5, several prior works have employed similar methodologies. For instance, the vision-based system in [54] aimed to assess Parkinsonian gait severity. However, their classifiers, trained on data with limited availability, despite achieving good correlation with clinician labels, faced challenges due to the small datasets. Similarly, Cho et al. [13] were limited by a dataset comprising only seven Parkinson’s patients and seven healthy controls.

Our study offers several advancements. First, we propose a novel gait representation method that allows us to leverage publicly available camera-based gait data from the INIT database. This approach captures detailed body movement information, akin to sensor-based methods, but with potentially broader accessibility and cost-effectiveness. Additionally, we employ an advanced deep learning method for multi-class gait classification, making it suitable for real-time implementation—a feature often absent in existing approaches. Overall, our work contributes a fresh perspective to gait analysis, addressing limitations in the current literature through innovative solutions.

In our study, we utilized a 1D CNN structure for its ability to effectively capture temporal information alongside main body movement features from silhouette binary images. Unlike 2D CNNs, which excel at learning contextual representations from 2D images, such as identifying key objects, our task presents challenges in identifying moving parts within a single silhouette.

One common approach to addressing this challenge is the generation of gait energy images (GEIs) by averaging energy across multiple sequences representing a gait cycle. However, we hypothesize that this method may not fully capture the available gait pattern information. Hence, we introduce a new approach to generate temporal information, incorporating body boundary angles and distances to create sinogram signals. Our choice to employ a 1D CNN allows us to effectively learn features by incorporating temporal representations from adjacent sequences.

To provide comparative insights, we implemented 2D CNN and 2D Residual Network (ResNet50) architectures using GEIs and compared their performance against our proposed 1D CNN using sinogram signals, utilizing identical training and testing splits. The results, presented in Table 6, demonstrate the superiority of our proposed 1D CNN approach over the 2D networks (CNN and ResNet50) for our specific task. This comparative analysis illustrates the efficiency of deep learning networks in capturing robust and prominent representations from the input data. Our method, which generates 1D sinogram signals, proves its capability compared to gait energy maps by considering calculations of angles and distances between all body boundary points and the centroid. This approach focuses more on body part movements, thereby enhancing gait impairment analysis.

## 5. Discussion

In this study, the following classification problems were tackled: the binary classification of NM versus FB, the classification of NM, FB, LL, and RL, and the classification of NM, FB, LL, RL, LA-0, and RA-0. The dataset comprised eight subjects for training and two subjects for testing for each anomaly; each of them had two sequences with a total number of frames of 818, 1292, 1258, 843, 842, and 2616 for NM, LL, RL, LA-0, RA-0, and FB, respectively. In addition, the effect of having different numbers of inputs was assessed: 1, 10, 20, and 30. Hence, testing the network with 20 and 30 images for the four-class and six-class problems, respectively, is recommended to attain the highest F1-score. It is worth noting that no misclassifications were encountered for FB when using image-level evaluation, and no misclassifications were encountered for RA-0, RL, LL, and FB when using subject-based evaluation. However, it is important to mention that the test set for subject-level evaluation was small: only two sequences for two subjects, which might be risky to generalize.

The use of 20 and 30 silhouette frames in our framework is considered sufficient to provide the necessary information for assisted knowledge learning. For instance, in [43], the number of silhouette frames within each gait cycle is estimated to be around 25 to 35 frames for normal motion styles. While this number may vary and increase for full-body gait patterns due to slower body movements, our experimental results indicate that abnormal full-body cases were the easiest class to classify compared to other gait patterns. Our study aimed to determine the optimal number of frames (k = 1, 10, 20, and 30) required to capture temporal information from adjacent sequences and identify the location of gait patterns. Increasing the number of frames would result in a heavier network architecture with more training parameters, as each frame is processed by a convolution layer to extract features before being incorporated with others. Although this may improve the performance to some extent, it could also lead to unfair comparisons with networks using fewer computations. Additionally, including more frames may introduce redundancy by incorporating information from multiple gait cycles of the same subject.

In the work of Verlekar et al. [43], where the input was the entire video sequence for testing purposes, they attained an accuracy of 98.8% at the subject level. Contrary to this, in this work, only 20 frames were sufficient to attain all the values of the F1-score, specificity, and accuracy at 100% at the subject level for the same four classes in INIT. In addition, when testing at the image level, the F1-score, specificity, and accuracy had values of 90.62%, 97.78%, and 90.66%, respectively. Regarding the six-class problem, the F1-score, specificity, and accuracy were 83.33%, 96.67%, and 83.33%, respectively, at the subject level. On the other hand, at the image level, the F1-score, specificity, and accuracy were 77.32%, 96.73%, and 77.7%, respectively.

The accuracy of the NM classification was 95.49%, 100%, 99%, and 99% in [13,43,44,46], respectively. The number of classes involved was 2, 4, 5, and 5 in [13,43,44,46], respectively. In this study, the NM classification exhibited an F1-score, specificity, and accuracy of 100% at the subject level, and 77.26%, 96.92%, and 93.95%, respectively, at the image level.

To the best of our knowledge, neither classifying into six classes nor image-level testing have been performed before. Although Albuquerque et al. [46] made their classification based on nine key frames, they needed the entire video sequence to detect those key frames in the first place.

## 6. Conclusions

The implementation of automated recognition algorithms for analyzing human gait patterns offers numerous valuable opportunities. The proposed method enables a quantitative and non-invasive assessment of locomotion by comparing observed locomotor function with a standard gait pattern often observed in individuals in good health. This work presents a novel pipeline for the examination of gait disability, which encompasses the following stages: The generation of silhouette sinogram signals is achieved by the utilization of a unique method based on angular centroid distance. Following that, a 1D CNN model is constructed and trained by incorporating shared information across multiple silhouette signals via assisted knowledge learning. This study utilized two separate evaluation methodologies. The initial approach entailed assessing every anticipated sinogram signal, which corresponds to an individual silhouette image, in comparison to its matching ground-truth label. The second evaluation system considered all the silhouette sinogram signals at the individual subject level, employing a majority vote approach. This study focused on three classification tasks: distinguishing between NM and FB in a binary classification issue, classifying NM, FB, LL, and RL, and classifying NM, FB, LL, RL, LA-0, and RA-0. Furthermore, an investigation was conducted to examine the effects of varying the quantity of signals or images inputted into the network. Specifically, this study explored scenarios where k equaled 1, 10, 20, and 30. According to the attained findings, it was seen that there were no instances of misclassifications in relation to FB deformation across all the values of n and k. This was particularly evident in cases when subjects exhibited pronounced impairment in their gait cycle, characterized by the presence of shortened steps, subtle foot–ground contact, and flexion in the knees, chest, and head.

Quantitative gait analysis is largely used in research institutes and not in therapeutic settings. Traditional gait laboratories’ high costs/cumbersome equipment, complex protocols, and data management/analysis, as well as clinical teams’ variable training/experience and preferences are the main reasons. Clinics use observational gait and qualitative measures most often, even if quantifying gait is less beneficial. It appears that our gait representation method has never been published. Similar to sensor-based approaches, low-cost camera-based gait data collecting can offer exact body component movement data. Real-time implementation is possible due to low processing complexity.

## Figures and Tables

**Figure 1 bioengineering-11-00477-f001:**
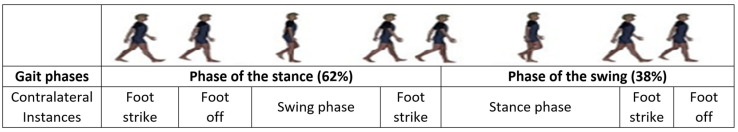
An example for a representation of a series of standardized binary silhouettes.

**Figure 2 bioengineering-11-00477-f002:**
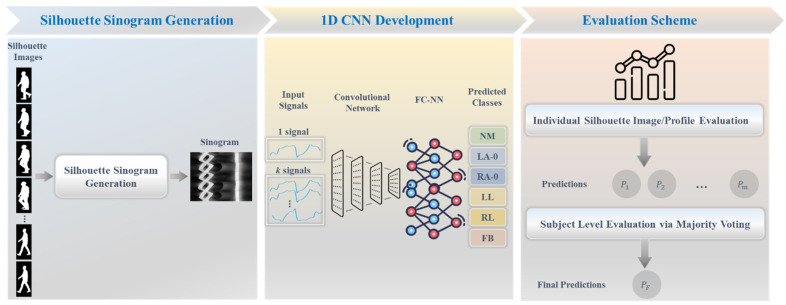
Overview diagram of the proposed framework, including silhouette sinogram generation from silhouette images, development of 1D CNN, and two evaluation schemes.

**Figure 3 bioengineering-11-00477-f003:**
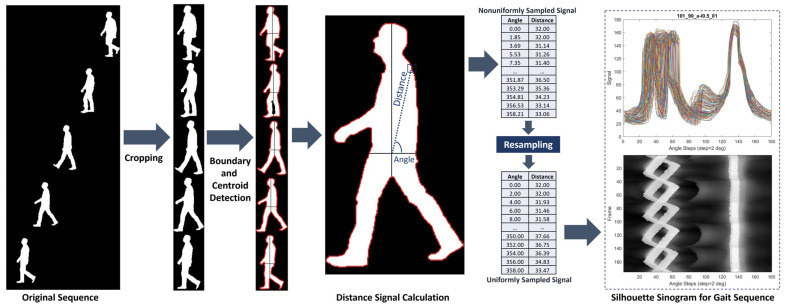
Schematic diagram of the generation process of the silhouette sinogram.

**Figure 4 bioengineering-11-00477-f004:**
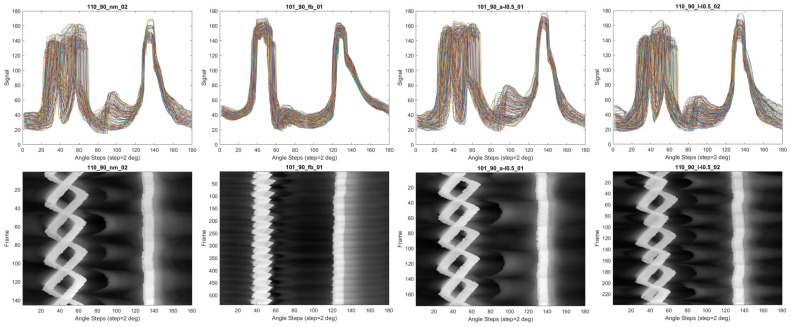
Silhouette sinogram signals for different gait patterns. Examples of normal (NM), full-body severe gait impairment (FB), motionlessness in the left arm, and left leg are shown from left-to-right, respectively.

**Figure 5 bioengineering-11-00477-f005:**
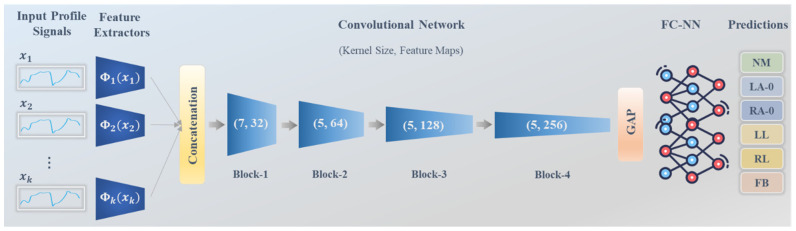
Schematic diagram of the proposed 1D CNN that involves multiple inputs serving as collaborative assisted knowledge learning.

**Figure 6 bioengineering-11-00477-f006:**
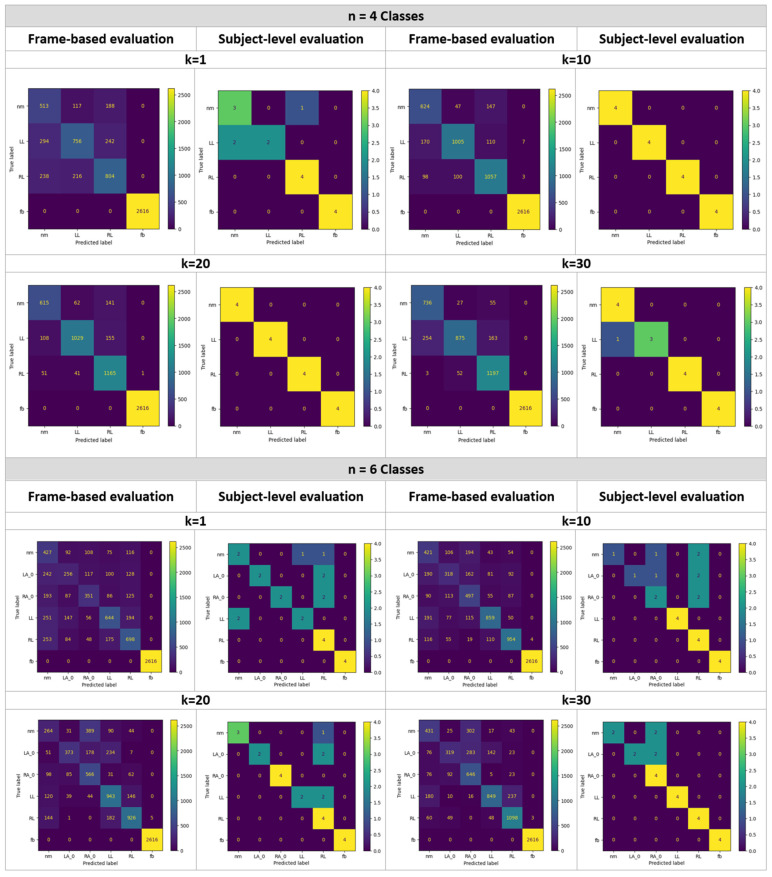
Confusion matrices of classifying n classes using k frames, where n = 4 (**top**) and n = 6 (**bottom**), and k = 1, 10, 20, and 30.

**Figure 7 bioengineering-11-00477-f007:**
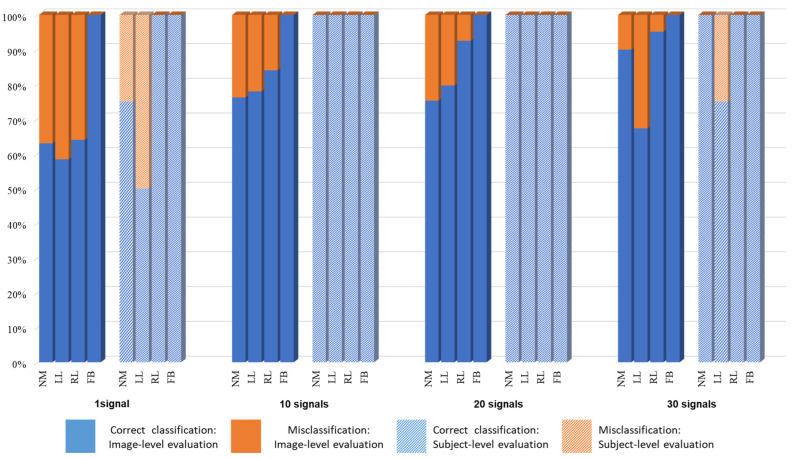
Correct vs. false classification for n = 4.

**Figure 8 bioengineering-11-00477-f008:**
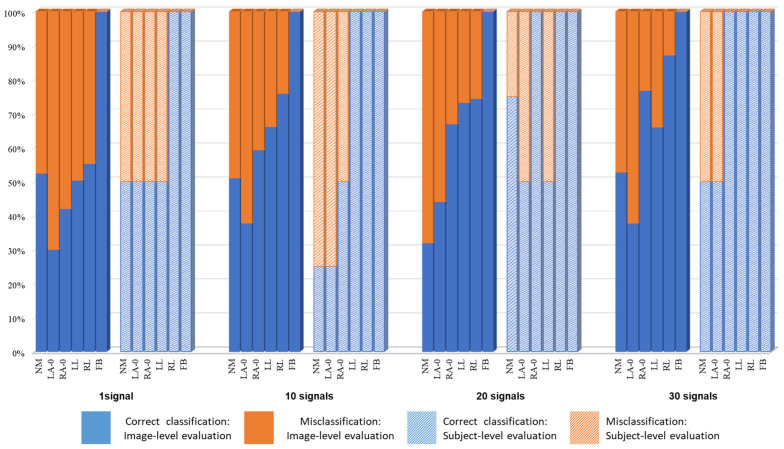
Correct vs. false classification for n = 6.

**Figure 9 bioengineering-11-00477-f009:**
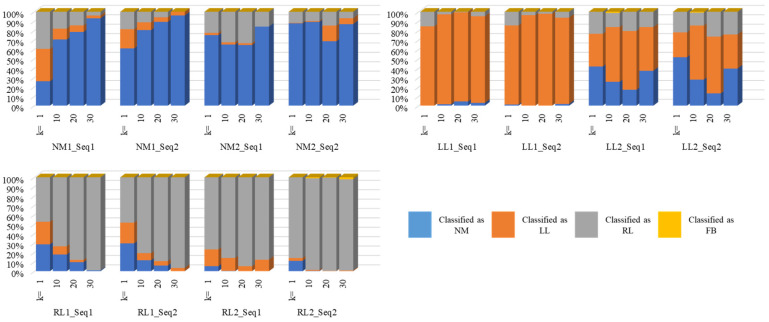
Four-class prediction labels for the test sets.

**Figure 10 bioengineering-11-00477-f010:**
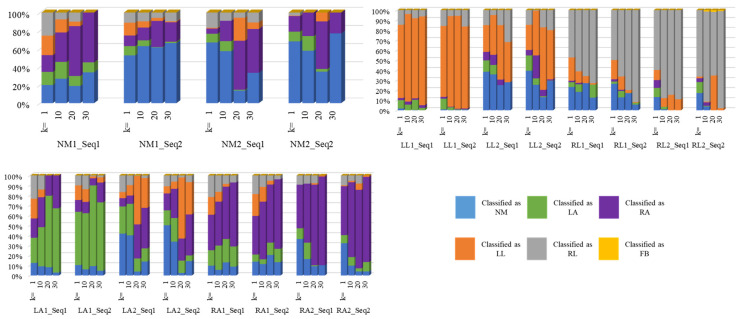
Six-class prediction labels for the test set.

**Table 1 bioengineering-11-00477-t001:** Description of INIT Gait database walking patterns.

Gait Style	Description
Normal (NM)	It depicts the normal gait pattern of a healthy person, which is also referred to as neutral or regular appearance in the database.
Abnormal right leg (RL)	A pattern of walking in which the right leg takes steps that are approximately one half as short as the ones taken by the left leg.
Abnormal left leg (LL)	A pattern of walking in which the left leg takes steps that are approximately one half as short as the steps taken by the right leg.
Abnormal full body (FB)	A damaged gait pattern in which the complete body shows a multitude of abnormal gait symptoms, including the following: participants walk slowly, bending their knees, and taking extremely tiny steps, barely lifting their feet off the ground (shuffling gait).
Abnormal half right arm (RA-0.5)	A walking pattern in which the right arm swings about one half as far as the left arm does.
Abnormal half left arm (LA-0.5)	A walking pattern in which the left arm swings about one half as far as the right arm does.
Abnormal right arm (RA)	A walking pattern in which the right arm does not swing at all while walking.
Abnormal left arm (LA)	A walking pattern in which the left arm does not swing at all while walking.

**Table 2 bioengineering-11-00477-t002:** Classification results of two evaluation schemes (signal-based and subject-based) with different number of prior frames (k = 1, 10, 20, 30) and two types of gait patterns (n = 4 and 6).

	Evaluation Based on a Signal/Image							
Classes	n = 4				n = 6			
Frames/signals	k = 1	k = 10	k = 20	k = 30	k = 1	k = 10	k = 20	k = 30
Sensitivity	78.36	88.60	90.66	90.64	65.09	73.91	74.17	77.70
Specificity	95.15	97.34	97.78	97.91	94.76	96.24	96.01	96.73
Accuracy	78.36	88.60	90.66	90.64	65.09	73.87	74.17	77.70
F1-score	78.52	88.65	90.62	90.43	65.32	74.05	73.88	77.32
	**Evaluation Based on Majority Voting of a Sequence of a Subject**							
Classes	n = 4				n = 6			
Frames/signals	k = 1	k = 10	k = 20	k = 30	k = 1	k = 10	k = 20	k = 30
Sensitivity	81.25	100	100	93.75	66.67	66.67	77.5	83.33
Specificity	93.75	100	100	97.92	93.33	93.33	95.24	96.67
Accuracy	81.25	100	100	93.75	66.67	66.67	79.17	83.33
F1-score	80.56	100	100	93.65	67	64.52	76.66	83.33

**Table 3 bioengineering-11-00477-t003:** Evaluation metrics per class for the suggested *k* values.

	Evaluation Based on a Signal/Image									
	NM		LL		RL		FB		LA	RA
Classes	n = 4	n = 6	n = 4	n = 6	n = 4	n = 6	n = 4	n = 6	n = 6	n = 6
Frames/signals	k = 20	k = 30	k = 20	k = 30	k = 20	k = 30	k = 20	k = 30	k = 30	k = 30
Sensitivity	75.18	52.69	79.64	65.71	92.61	87.28	100	100	37.84	76.72
Specificity	96.92	94.28	97.8	96.68	93.74	94.91	99.97	99.94	97.42	91.2
Accuracy	93.95	89.84	93.88	91.46	93.5	93.66	99.98	99.96	90.87	89.61
F1-score	77.26	52.53	84.9	72.16	85.69	81.88	99.98	99.94	47.68	61.85
	**Evaluation Based on Majority Voting of a Sequence of a Subject**									
Frames/signals	k = 20	k = 30	k = 20	k = 30	k = 20	k = 30	k = 20	k = 30	k = 30	k = 30
Sensitivity	100	50	100	100	100	100	100	100	50	100
Specificity	100	100	100	100	100	100	100	100	100	80
Accuracy	100	91.67	100	100	100	100	100	100	91.67	83.33
F1-score	100	66.67	100	100	100	100	100	100	66.67	66.67

**Table 4 bioengineering-11-00477-t004:** Samples of the input signals and their corresponding predictions.

Correctly classified FB using n = 2, 4, and 6 for any k value. Blue corresponds to predicting FB.	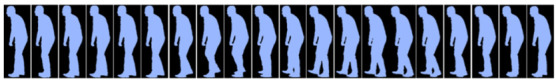
A part of an NM sequence; white, red, and green corresponds to NM, LL, and RL predictions, respectively. n = 4, k = 1.	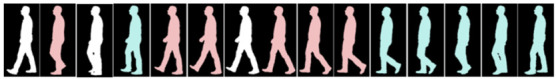
Same part as the row above; white and green corresponds to NM and RL predictions, respectively. n = 4, k = 10, 20.	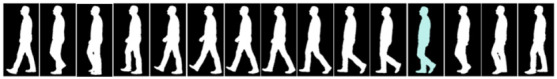
Same part as the row above; white and green corresponds to NM and RL predictions, respectively. n = 4, k = 30, and for n = 2.	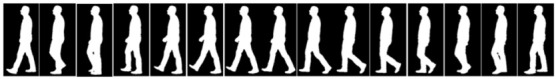
A part of an LL sequence; white, red, and green corresponds to NM, LL, and RL predictions, respectively. n = 4, k = 1.	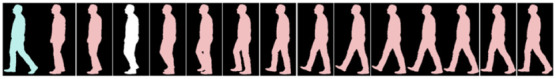
Same part as the row above; red corresponds to LL predictions. n = 4, k = 10, 20, and 30.	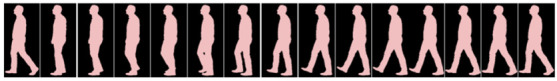

**Table 5 bioengineering-11-00477-t005:** Comparison of used dataset, methodology, and achieved accuracy.

Study	Data	Approach	Accuracy
Chavez et al. [54]	Private dataset (Parkinsonian gait)	Classical machine learning methods (RFC and XGBoost)	50% (Normal vs. abnormal gait)
Cho et al. [13]	Private dataset (Parkinsonian gait)	Classical machine learning methods (hybrid PCA and LDA)	77% (Normal vs. abnormal gait)
Proposed study	Public dataset (INIT database [50])	Deep learning approach (1D CNN)	90% (4 classes of gait) 77% (6 classes of gait)

RFC: random forest classifier, XGBoost: extreme gradient boosting, PCA: principal component analysis, LDA: linear discriminant analysis.

**Table 6 bioengineering-11-00477-t006:** Comparison of the proposed 1D CNN that uses the generated 1D sinogram signals against 2D CNN and ResNet50 models that use 2D gait energy images (GEIs).

Methods	Number of Classes (n = 4)				Number of Classes (n = 6)			
	Sensitivity	Specificity	Accuracy	F1-Score	Sensitivity	Specificity	Accuracy	F1-Score
2D CNN	82.50	94.83	82.50	82.35	70.94	89.90	70.94	70.25
2D ResNet50	86.25	91.38	86.25	86.24	67.08	85.86	67.52	67.08
Proposed 1D-CNN (k = 20)	90.66	97.78	90.66	90.62	74.17	96.01	74.17	73.88
Proposed 1D-CNN (k = 30)	90.64	97.91	90.64	90.43	77.70	96.73	77.70	77.32

## Data Availability

The authors confirm that the data used in this study are publicly available at https://www.vision.uji.es/gaitDB/ (accessed on 10 February2023).

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
