# Peer review of "Gait Impairment Analysis Using Silhouette Sinogram Signals and Assisted Knowledge Learning"

_bioengineering, 2024, doi:10.3390/bioengineering11050477_

Round 1
Reviewer 1 Report
Comments and Suggestions for Authors
I have read the paper completely. The authors firstly generated a novel one-dimensional representation of each silhouette image, termed silhouette sinogram, by computing the distance and angle between centroid and each detected boundary points. This process enables to effectively utilizes relative variations of motion at different angles to detect gait patterns. Also, one-dimensional convolutional neural network (1D CNN) model is developed and trained by incorporating the consecutive silhouette sinogram signals of silhouette frames to capture spatiotemporal information via assisted knowledge learning by the authors. This process allows the network to capture a broader context and temporal dependencies within the gait cycle, enabling more accurate diagnosis of gait abnormalities. This study conducts training and evaluation utilizing the publicly accessible INIT GAIT Database. At the last step of the submitted paper, two evaluation schemes are employed: one leveraging individual silhouette frames and the other operating at the subject level, utilizing a majority voting technique.
Various studies about the paper's topics have been carried out recently. Therefore the study is up to date. After review, I think that the idea of this paper’s results are new correct and scientificaly sound. The paper is well written and the results are highlighted in this field.
Reviewer 2 Report
Comments and Suggestions for Authors
Authors propose a novel gait impairment analysis method based on the silhouette sinogram signals and assisted knowledge learning from video-recorded images in an efficient manner. To improve the quality of this manuscript, my comments and suggestions are listed as follows.
1. Major issues
(1) For Section 3.2, authors are suggested to describe their method with detailed steps. They can list the steps of their method instead of a long paragraph. Meanwhile, authors are suggested to describe their method with the math symbols and equations as more as possible.
(2) Authors state that they design a 1D CNN structure. What are its advantages over the traditional 2D CNNs? Authors should state or make comparisons in the manuscript.
(3) Authors should make the comparative experiments and analyses with the similar methods.
(4) Authors only use 20 and 30 images for test. Why do authors not use more test images? Does the accuracy can be improved by more test images?
2. Minor issues
(1) The title is a little long. Authors can try to make it concise. What about the following title?
“Gait Impairment Analysis Using Silhouette Sinogram Signals and Assisted Knowledge Learning”
(2) For the Section Abstract, the research background and method description parts have too much trivial details. Authors should make them concise. Meanwhile, the conclusion part of Section Abstract is a little weak or unclear.
(3) For the keywords, “gait analysis; gait impairment; gait pathology classification” can be optimized. These keywords have the same word “gait”.
(4) The paper architecture should be given as the last paragraph of Section Introduction. It tells reader the content in each section.
(5) There are some typos and English grammar errors. Authors should carefully polish their manuscript. For example, the Eq. (2) should appear in the 429th line.
(6) Most of the references are very old. Authors should make a new literature survey and update the references.
(7) Authors are suggested to offer the source of the databases, such as giving the URL in this manuscript.
(8) The section number used in this manuscript is not standard. For example, in Subsection 3.2.1. Generation of Silhouette Sinogram Signals, “1. Image Cropping” should be “(1) Image Cropping”.

There are some typos and English grammar errors.
Reviewer 3 Report
Comments and Suggestions for Authors
The implementation of automated recognition algorithms for analyzing human gait patterns offers numerous valuable opportunities. The proposed method enables a quantitative and non-invasive assessment of locomotion by the comparison of observed locomotor function with a standard gait pattern often observed in individuals with good health. Article Overall, it is quite interesting, but it needs to be improved in terms of literature review, innovation, language expression, etc. I agree its publication after revision according to my suggestions. Here are some specific suggestions:
1. The authors develop a 1DCNN model capable of integrating multiple silhouette sinogram signals and training them together to enhance the network's understanding of gait abnormalities through assisted knowledge learning. It is worth mentioning that 1DCNN is a widely used deep learning method, this article Just application, not innovative
2. The authors analyze the influence of varying 𝑘 assisted silhouette sinogram signals, used as a prior knowledge and fed into the network (𝑘 = 1, 10, 20, and 30). What is the reason for these values of k?
3. Some method reviews on deep learning in the literature review of the article are relatively outdated. It is recommended that the authors analyze some recent learning-related work, such as multi-hop graph pooling adversarial network for cross-domain remaining useful life prediction: a distributed federated learning perspective,lifetime extension approach based on levenberg-marquardt neural network and power routing of dc-dc converters
4. The article lacks sufficient theoretical foundation and perfect formula expression. It is necessary to strengthen the theoretical foundation.
5. Some pictures in the article are not clear enough. It is recommended to modify the format of the pictures to increase the clarity of the pictures.
6. The comparative experiments in the article are not sufficient. It is recommended that the authors compare with some recent DL-based methods.
7. The method used in the article seems to already exist, so what is the innovation of the article?
Comments on the Quality of English LanguageN/A
Round 2
Reviewer 2 Report
Comments and Suggestions for Authors
Authors have carefully revised their manuscript according to my comments and suggestions. The quality of this manuscript is improved obviously. I am satisfied with their revisions. Therefore, I think that this manuscript can be accepted in the current version at present.
Reviewer 3 Report
Comments and Suggestions for Authors
Thanks the authors' revision. I accept its publication.
Comments on the Quality of English LanguageThanks the authors' revision. I accept its publication.